# Evaluating Eye Tracking Technology in Nursing Education: A Scoping Review on Medication Administration Training

**DOI:** 10.3390/nursrep15060185

**Published:** 2025-05-27

**Authors:** Jiranut Chaichana, Rob Eley, Chris Watling, Linda Ng

**Affiliations:** 1School of Nursing & Midwifery, University of Southern Queensland, 11 Salisbury Road, Ipswich, QLD 4305, Australia; jiranut.chaichana@unisq.edu.au; 2Princess Alexandra Hospital Southside Clinical Unit, Faculty of Health, Medicine and Behavioural Sciences, University of Queensland, 199 Ipswich Rd, Woolloongabba, QLD 4102, Australia; r.eley@uq.edu.au; 3School of Psychology, University of Southern Queensland, 11 Salisbury Road, Ipswich, QLD 4305, Australia; chris.watling@unisq.edu.au

**Keywords:** eye tracking technology, medication errors, medication training, nursing education, nursing students

## Abstract

**Background:** Eye tracking technology, when used in nursing, helps to reduce medication errors by analyzing eye movements. In education, it provides insights into student learning, cognitive load, and instructional design, allowing for more personalized learning. Despite challenges such as the need for technical expertise, privacy concerns, and cost, eye tracking offers real-time feedback that enhances both teaching and learning effectiveness. **Objectives**: To explore the current evidence on the application of eye tracking technology in training nursing students for drug administration. **Methods**: Eligible studies included peer-reviewed empirical papers, both qualitative and quantitative, and reports published in English. Excluded were studies involving Non-Eye Glass Tracking, non-nursing students, or non-English articles. Searches were conducted in nine databases. The risk of bias was assessed using the JBI SUMARI tool, and the results were synthesized narratively, presented with the PRISMA-P flow diagram. **Results**: From 739 studies, 10 focusing on medication training were identified. Eye tracking helped to reveal differences in visual focus between novice and expert nurses, with certain interventions shown to improve attention and concentration. **Conclusions**: Eye tracking has strong potential in nursing education, especially for improving attention and enhancing situational awareness in medication administration. However, limitations such as small sample sizes, technical barriers, and a lack of long-term data remain. Future research should address these gaps with larger, more diverse samples and extended follow-ups.

## 1. Introduction

The World Health Organization’s 2022 technical document on medical errors highlights the substantial economic burden these errors impose globally [1]. In Europe, the annual costs are estimated to range from EUR 4.5 billion to EUR 21.8 billion [2]. One country alone reported EUR 2 billion in medical errors, accounting for 3% of its public health expenditure. Additionally, errors in primary care that lead to hospitalization due to avoidable substance misuse cost approximately GBP 98.5 million annually, resulting in hundreds of deaths [3].

A study conducted in hospitals affiliated with Iran University of Medical Sciences which examined medication errors (MEs) among 503 nurses reported that on average, each nurse experienced precisely 6.27 MEs per month. The most common errors included prescribing medication at the wrong time (28.4%), administering too many medications simultaneously (22.4%), and failing to provide analgesics after surgery (15.3%). Factors such as work experience and secondary employment were reported to influence error rates. Major sources of error included high and irregular nurse turnover, changes in medication dosages due to multiple consultations, and managerial unresponsiveness to identified errors [4].

An integrative review of medication errors by nursing students during clinical practice identified six full-text articles and found that medication errors occurred at a rate of 1–6% [5]. These errors were attributed to factors related to the students themselves, their education, and their working environments. The types of errors included miscalculations in medication dosages, using the wrong patient’s name, administering the incorrect medication, and omitting doses.

Eye tracking technology records where healthcare professionals and students’ eyes move while undertaking an activity. There are various ways in which eye tracking is used in medical practice. It provides insight into the psychological processes occurring, and can be used to enhance both training and safety. In the context of healthcare education for medication administration, eye tracking is particularly valuable for understanding healthcare students’ learning strategies. It reveals how visual acuity develops and what can lead to errors during training. Eye tracking helps nursing students and healthcare professionals to better understand their visual attention and decision-making processes during these crucial tasks [6].

Eye tracking technology typically involves tools such as smart glasses that record visual attention [6,7]. To understand how to analyze eye movement, an Area of Interest (AOI) refers to a specific region in a visual display that researchers analyze to understand cognitive effort. It is sometimes called a Region of Interest (ROI), and is used to measure how people process information [7]. Eye tracking tools commonly used in nursing education include smart glasses and head-mounted devices that monitor gaze behavior in real time [7]. These devices collect data on visual metrics such as fixation duration, saccades (rapid eye movements), and scan paths to evaluate cognitive workload and attention patterns [6]. The Eye Movement Matrix measures key aspects such as fixation duration (how often a person looks back at the same spot after looking elsewhere), number of fixations (how often gaze returns to an area), and saccades (quick eye movements between fixations) [8]. These metrics help to analyze visual attention, mental workload, and task efficiency [9].

The visual observation method has contributed significantly to training, assessment, and feedback practices in the clinical setting by providing reliable quantitative data. For instance, fixation patterns can indicate where student nurses direct their attention and highlight attentional issues, such as information that is overlooked during procedures. Research has shown differences in gaze behavior between novice and expert nurses, with novice nurses taking longer to complete tasks and demonstrating lower injection accuracy compared to experts [10]. These data can be interpreted to assess clinical competence, provide training solutions, and support feedback concept processes to enhance the overall educational experience [11]. Additionally, qualitative data from eye tracking studies in medication training highlight the importance of understanding how trainees interact with medication information. For instance, a study by Andry [12] found that eye tracking combined with semi-directive interviews revealed insights into participants’ emotions and cognitive processes during medication training. This approach helped to identify areas where trainees struggled with information processing and provided a deeper understanding of their learning experiences. Such qualitative insights are invaluable for tailoring training programs to better meet the needs of learners and improve overall educational outcomes.

Furthermore, the aim of this scoping review is to explore the current evidence on the application of eye tracking technology in training nursing students for medication administration. This review seeks to understand how eye tracking technology is utilized in the training of nursing students for medication administration. It aims to evaluate the effectiveness of this technology in enhancing the training outcomes for nursing students. Additionally, the review identifies the challenges faced in the implementation of eye tracking technology in nursing education. This review specifically focuses on the use of Eye Glass-Based Tracking tools, such as smart glasses, which are wearable technologies capable of capturing real-time eye movements. Finally, it will assess the outcomes achieved using eye tracking technology in the training process.

By examining these aspects, the review intends to provide a comprehensive understanding of the role and impact of eye tracking technology in nursing education, particularly in the context of drug administration training. This understanding could inform future educational strategies and technological integrations in nursing curricula, ultimately aiming to improve the quality and effectiveness of nursing education.

## 2. Research Questions

This scoping review was guided by four research questions aimed at exploring the integration of eye tracking technology in nursing education, specifically for medication administration. Firstly, it examined the current applications of eye tracking technology in training nursing students, identifying various methodologies and contexts in which this technology is utilized. Secondly, the review assessed the effectiveness of eye tracking technology in enhancing the accuracy and efficiency of nursing students during medication administration, providing insights into its impact on performance. Thirdly, it investigated the primary challenges faced by both educators and students when incorporating eye tracking technology into the curriculum, highlighting barriers and potential solutions. Lastly, the review explored the short-term and long-term outcomes of using eye tracking technology in nursing training, evaluating its benefits and implications for future educational practices. Through these research questions, the review aimed to provide a comprehensive understanding of the role and impact of eye tracking technology in nursing education.

## 3. Materials and Methods

A protocol for this study has been registered in the Open Science Framework (https://doi.org/10.17605/OSF.IO/KR87A). The scoping review methods followed the Joanna Briggs Institute (JBI) Methodology for Scoping Reviews [13] and the PRISMA-ScR (Preferred Reporting Items for Systematic Reviews and Meta-Analyses extension for Scoping Reviews) framework [14], with the search strategy guided and developed by an academic librarian to ensure a rigorous and transparent process.

### 3.1. Eligibility Criteria

For this scoping review, inclusion and exclusion criteria were meticulously selected to ensure a focus on the most relevant and high-quality studies related to the use of eye tracking technology in nursing education. These criteria were designed to identify studies that were both reliable and aligned with the review’s aims objectives, thereby enhancing the validity and relevance of the findings (Table 1).

### 3.2. Search Strategy

Initially, a targeted search was conducted in PubMed and MEDLINE (Ovid) to identify articles primarily focused on Eye Glass-Based Tracking. The selected texts were then analyzed to identify key index terms within this domain. With additional support from a medical librarian, an initial search strategy was developed using the identified key terms related to ‘Eye Glass-Based Tracking,’ ‘nursing students,’ and ’medication’. This search strategy was adapted for use in multiple databases: CINAHL, Embase, Google Scholar, Ovid, ProQuest One Academic, PubMed, ResearchGate, ScienceDirect, and Scopus. All databases were last searched on 20 November 2024.

### 3.3. Study Selection

The selection of papers for this study was conducted with strict adherence to the inclusion and exclusion criteria outlined in Table 1. To ensure both thoroughness and accuracy, the selection process was carried out in two distinct stages.

**Table 1 nursrep-15-00185-t001:** Inclusion and exclusion criteria.

Inclusion Criteria	Exclusion Criteria
Type of papers: peer-reviewed empirical papers, qualitative papers, quantitative papers, theses, government reports, conference proceedings, organizational reports, business reports, and newspaper articles	Articles reporting synthesis of primary research (e.g., systematic, scoping, or narrative reviews)
Language: English	Papers in languages other than English
Population of focus: nursing students	Non-nursing students, such as public health students and physiotherapy students
Primary focus: Eye Glass-Based Tracking (studies using smart glasses or augmented-reality glasses with integrated eye tracking)	Another focus, such as Non-Eye Glass Tracking (research using screen-based, head-mounted, or stationary eye trackers) and Non-Gaze Wearables (studies involving wearables without eye tracking functionality)
Setting: any setting where tertiary nursing education or training is conducted, including universities, colleges, and hospital-based nursing programs	Another setting, such as a high school or school

#### 3.3.1. Initial Screening

Search records were imported into EndNote 21 (Clarivate Analytics, Philadelphia, PA) to facilitate the removal of duplicates and the identification of potentially relevant studies. This process combined automated functions with manual verification to ensure accuracy. The study screening process was then managed using JBI SUMARI (https://sumari.jbi.global) (accessed on 26 Jan 2025), which supported the systematic review workflow and documentation.

Two reviewers (J.C. and R.E.) independently screened the titles and abstracts of all identified studies to assess their relevance based on predefined inclusion and exclusion criteria. The inclusion criteria required studies to meet the following requirements: (1) involve the use of Eye Glass-Based Tracking or wearable eye tracking devices; (2) focus on nursing students or nursing education; and (3) address medication administration, medication safety, or related clinical decision-making processes. Studies were excluded if they had the following characteristics: (1) did not involve nursing students as participants; (2) focused solely on non-wearable eye tracking technologies; (3) were not published in English; or (4) were conference abstracts, editorials, or opinion pieces without primary data.

Any discrepancies between the reviewers were discussed and resolved through consensus. In cases where consensus could not be reached, a third reviewer was consulted to make the final decision. This dual-reviewer approach helped to minimize selection bias and enhance the reliability of the screening process.

#### 3.3.2. Title and Abstract Screening

Two reviewers (J.C. and R.E.) independently screened the titles and abstracts of the identified studies to select a subset for further review.

#### 3.3.3. Full-Text Evaluation

The complete texts of the selected studies were evaluated by the two independent reviewers (J.C. and L.N.) based on the established inclusion and exclusion criteria. During this full-text screening process, both reviewers provided reasons for excluding sources of evidence that did not meet the specified criteria for the scoping review [15]. Any discordance was discussed and resolved.

The findings from the search and the study inclusion process have been thoroughly documented in the final scoping review. The results are presented using the Preferred Reporting Items for Systematic Reviews and Meta-Analyses Protocols (PRISMA-P) flow diagram [14] (Figure 1).

### 3.4. Data Analysis and Presentation

The data were analyzed using descriptive statistics and grouped according to the research questions and objectives. Key variables included ‘Study/Year’ for relevance, ‘Origin’ for geographical context, ‘Research Design’ for methodology, and ‘Aims/Purpose/Objectives’ for central goals. ‘Population and Sample Size’ provided insights into participant representativeness, while ‘Data Collection Method’ outlined the techniques used. ‘Main Findings’ highlighted key results, with ‘Eye Tracking Outcomes’ focusing on visual attention, and ‘Other Outcomes’ covering secondary findings. The ‘Recommendation’ variable included suggested actions. Assumptions and simplifications were made by aligning variables with the study’s objectives and focusing on broad categories, such as sample size, rather than detailed participant demographics. A summary of the included data is given in Table 2.

## 4. Results

The database searches found 739 studies, of which 28 were duplicates. After checking the remaining 711 studies based on their titles and abstracts, 12 studies were reviewed in full text. Two were excluded as they did not meet the inclusion criteria, and ten studies were included in this review.

### 4.1. Characteristics of the Included Studies

The studies originated from four of countries: Japan (*n* = 4), the United States (*n* = 3), Canada (*n* = 2), and Spain (*n* = 1). Study methodologies included quantitative studies (*n* = 7), randomized studies (*n* = 2), and observational studies (*n* = 1) (Figure 2). Seven studies focused solely on nursing students, two studies included a mix of nursing students and other healthcare professionals (e.g., educators, medical experts, physicians), and one study involved a mix of nursing students and nurses. No students of other healthcare professions were identified in the literature. All the studies took place in simulation laboratories.

### 4.2. The Use of Eye Tracking Technology in Medication Administration Training

The scoping review highlights the use of eye tracking technology in nursing education, particularly in medication administration training. Several studies investigated visual attention, which was assessed via fixation, gaze patterns, gaze durations, and transitions during tasks such as intravenous injection, subcutaneous injections, and infusion pump operation [17,19]. Eye tracking revealed differences in gaze behavior between novice and expert nurses, where novice nurses were more likely to take longer to complete tasks and make less accurate injections than experts. These findings demonstrate the role of eye tracking in skill development by helping trainees to improve their accuracy and visual focus during critical nursing procedures [18]. They spent more time looking at the syringe and needle, while experts focused more on the injection site. Novices also had more frequent gaze shifts between areas of interest, showing that they had less control over their attention compared to experts. These findings aid in understanding differences in attentional focus during medication tasks between novice and expert nurses [18,22]. Interventions such as Quiet Eye (QE) training, which teaches individuals to keep their eyes focused on key points before making a movement, help to improve accuracy [10]. Similarly, the Stay S.A.F.E. strategy is a step-by-step method that helps nurses to stay focused on their tasks while handling interruptions. Both approaches have been shown to enhance visual focus during tasks, even under a mental workload or interruptions [9]. The-Eye technology, as a debriefing technique in simulation-based teaching, has been studied alongside verbal debriefing and a combined approach. Research has shown that all three methods improve post-test performance compared to pre-test evaluations. However, eye tracking feedback alone has demonstrated a significant impact on learning outcomes, particularly in enhancing patient safety practices [21].

### 4.3. Eye Tracking Technology and Reduction of Medication Errors

Eye tracking technology has been implemented in various nursing education interventions with the aim to identify and reduce medication errors. Key interventions include monitoring visual focus during medication administration, comparing gaze patterns of nursing students and nurses during injection procedures [16], and using the Stay S.A.F.E. strategy to improve focus during interruptions [9]. Findings show that experienced nurses have more focused and efficient gaze patterns, helping them to complete tasks correctly and quickly. On the other hand, nursing students often have scattered attention, which causes them to overlook important steps in the process. As a result, they miss critical actions, increasing the chance of making mistakes [19,22]. These results demonstrate that eye tracking technology helps to identify patterns of inattention, offering valuable insights that can be used to design targeted training to improve focus and reduce errors [19,20,22].

### 4.4. Knowledge Gaps and the Impact of Eye Tracking on Medication Safety Training

The scoping review highlights the need to improve nursing students’ knowledge, particularly in medication classifications and critical tasks, to reduce errors [9,20,22]. A key finding is that students who struggle to identify medication errors often lack knowledge of drug classifications [20] and are more easily distracted, focusing less on critical tasks compared to experienced nurses. Eye tracking technology has been identified as a tool that can help to pinpoint where students struggle with attention and focus [22], particularly in tasks such as intravenous injections [17,19].

These findings show that eye tracking plays a meaningful role in mistake prevention by identifying visual attention lapses and revealing patterns of distraction or uncertainty during critical tasks. For example, Amster et al. [20] reported that students who missed allergy-related errors did not differ in terms of gaze patterns from those who caught them, suggesting gaps in pharmacological knowledge that eye tracking could help to uncover and address.

To address these challenges, recommendations include incorporating simulations, self-learning systems, and more focused training to enhance medication safety education [19]. Additionally, strategies such as the Stay S.A.F.E. program could assist students in managing distractions and maintaining focus [9]. However, the review also emphasizes the need for long-term studies to assess the real-world effectiveness of eye tracking training [18] and its impact in clinical practice. Current studies face limitations such as small sample sizes and technical challenges, necessitating further research with larger participant groups and improved technology [10,16,17,19,20].

## 5. Discussion

This review presents the different ways in which eye tracking technology is used in nursing education and training related to medication administration. The reviewed studies focus on training and analyzing gaze patterns during important nursing tasks, providing valuable insights into skill development, attention control, and mistake prevention.

### 5.1. Improving Skill Development and Training

Some studies used eye tracking technology to observe where the participants focused their gaze during tasks such as giving intravenous injections and subcutaneous injections [17,19]. The results showed obvious differences between novice and expert nurses. Experts focused on key areas, such as clinically relevant areas (i.e., the patient’s arm and vein), while novices’ gaze patterns were more widespread and less efficient. This suggests that nursing students can gain benefits from training that helps them to focus on the most important points during these procedures [10]. This training could improve succession rates, reduce mistakes, and make procedures faster.

These results illustrate how eye tracking supports skill development by showing students how to align their visual attention with expert performance during clinical tasks.

### 5.2. Improving Awareness and Preventing Mistakes

Building upon the skill-based outcomes discussed previously, this section shifts focus to how eye tracking technology assists in identifying and addressing attentional failures that contribute to medication errors. Eye tracking technology has highlighted attention management as a challenge for nursing students [8,20]. Studies indicate that students often miss critical steps, such as adjusting drip speeds, due to divided attention [17]. Additionally, distracted gaze behavior during multitasking has been linked to slower task completion and increased error risk [22]. These findings suggest that improving attention control enhances medication administration accuracy and overall patient safety [21]. The correlation between gaze behavior and medication errors lies in how attention control and visual focus can influence procedural accuracy. Eye tracking supports interventions that aim to reduce these errors, rather than serving as a direct solution. Furthermore, the results indicate that eye tracking could help to address the areas where nursing students struggle and could be used to create specified training to improve their task awareness, especially under pressurized situations.

### 5.3. Understanding How Eye Tracking Works

Eye tracking technology helps researchers and teaching staff to understand how nursing students focus their attention during medication administration procedures. In nursing education, eye tracking is used to study gaze patterns during tasks such as medication administration and intravenous injections [16,17]. The Eye Movement Matrix tracks important factors such as fixation duration, the number of fixations, and saccades [8]. These measurements help to analyze visual attention, mental workload, and task efficiency [9]. Researchers identify Areas of Interest (AOIs), such as the syringe, injection site, and patient information, to assess attentional focus [16,18]. Studies show that expert nurses have more efficient gaze patterns, spending more time on critical areas and making fewer unnecessary eye movements compared to students [18,19]. Data collection involves wearable eye trackers, which record gaze positions and transitions between AOIs [17]. These data are analyzed to identify patterns in visual attention, helping educators to improve training methods [17,18]. Moreover, eye tracking is a useful tool in assessing cognitive workload, indicating that experienced nurses encounter lower mental strain when carrying out complex tasks. By incorporating eye tracking into nursing education, researchers can formulate targeted strategies to minimize errors and improve patient safety [9].

### 5.4. Eye Tracking as Feedback in Nursing Education

Some studies have explored how eye tracking technology could be used as a debriefing method for students. For example, when eye tracking data were combined with traditional verbal feedback, students could see where they focused their attention and how it affected their safety practices. This helped them to improve their performance in later evaluations. In other cases, eye tracking itself was helpful, especially in tasks that required visual focus, such as checking patient data. This suggests that eye tracking technology can be a useful tool for providing feedback in simulation-based training, either by itself or alongside other forms of feedback. It helps students to improve their observation skills and focus on patient safety [21].

### 5.5. Recommendations and Future Research

These findings show that eye tracking technology can play an important role in nursing education by helping to close the gap between novice and expert performance with medication procedures. Some recommendations include creating self-learning systems where students can watch the gaze patterns of expert nurses, allowing them to learn by observing [19]. Additionally, students should receive more education on medication and commonly used drugs to reduce the risk of medication errors. Eye tracking technology can be used to identify and address knowledge gaps in this area [20]. Furthermore, simulation-based training programs should be developed to enhance nursing skills using eye tracking feedback. These programs can improve not only technical skills [19], but also situational awareness [17] and the ability to manage a mental workload [22].

Furthermore, these skills require repetitive practice until the brain develops procedural memory, or ‘rote learning,’ which helps nursing students to perform critical tasks automatically and safely. This cannot be stressed enough. Studies in nursing education confirm that deliberate and repeated practice improves performance and reduces errors [11].

Future research should explore the long-term effects of eye tracking-based interventions on clinical outcomes. Although the reviewed studies indicate promising outcomes in enhancing visual attention, skill acquisition, and error reduction, the current evidence remains preliminary. Eye tracking technology may be best introduced through pilot programs or targeted training modules to assess feasibility and educational impact. Broader integration into nursing education curricula should only be considered following validation through large-scale, longitudinal studies across diverse educational settings [10,18,19,20].

It is important to note that while eye tracking technology enhances skill development through feedback on visual attention, it is not intended to replace foundational pharmacological knowledge. Rather, it serves as a complementary tool to strengthen learning and promote safer medication administration practices [11].

### 5.6. Limitations

This scoping review was limited to studies published in English, which may have caused bias by excluding relevant research in other languages. Additionally, the review focused only on nursing students, so the results may not apply to other professional students. The choice of databases used in the search may have affected the range of studies included, and using different search terms could have led to the discovery of other relevant studies. There is also potential for publication bias, as studies with positive findings are more likely to be published, and the exclusion of non-English literature may have limited the comprehensiveness of the evidence base.

## 6. Conclusions

This study shows that eye tracking technology can improve nursing student education, especially in teaching medication administration skills. The studies included in this review highlight how eye tracking technology helps to identify differences in visual focus between novice and experienced nurses. The review identified several training methods, such as Quiet Eye training and the Stay S.A.F.E. strategy, that have been shown to improve concentration, decrease errors, and increase task awareness. However, there are still areas for advancement. Many studies had small sample sizes, faced technical issues, and did not include follow-up in long-term research. These factors make it difficult to apply the findings broadly. Future research should focus on larger samples of participants, including more healthcare profession students. In summary, eye tracking technology is a promising tool for improving nursing students’ skills in medication administration, and shows utility as a training tool to augment existing teaching practices in nurse skill acquisition.

## Figures and Tables

**Figure 1 nursrep-15-00185-f001:**
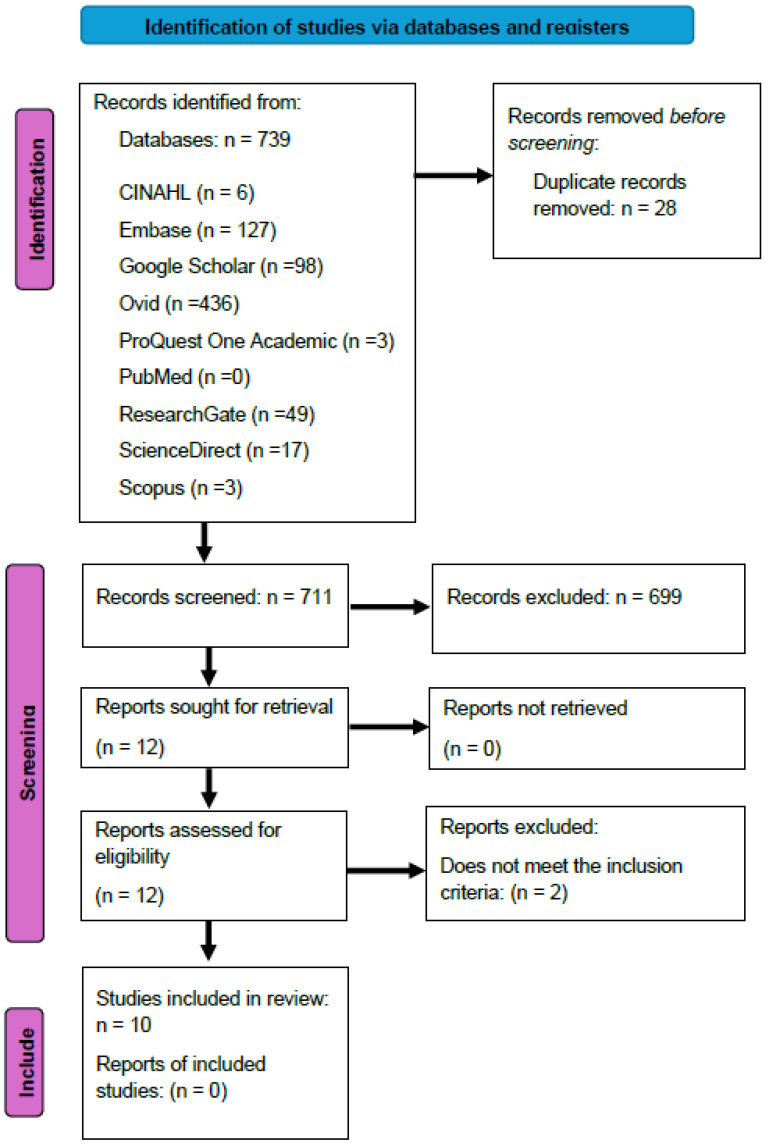
PRISMA flow diagram of selection process.

**Figure 2 nursrep-15-00185-f002:**
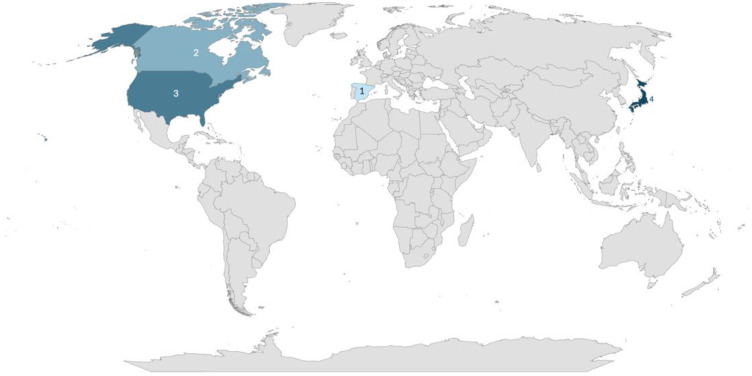
Study distribution map.

**Table 2 nursrep-15-00185-t002:** Study characteristics: year, origin, research design, aims/purpose/objectives, main findings, and recommendations (N = 10).

Study/Year/ Origin/Research Design	Aims/Purpose/Objectives	Main Findings	Recommendations
Parker et al. (2024)/Canada/a randomized controlled trial (RCT) [10]	To examine the effectiveness of Quiet Eye (QE) training compared with traditional technical (TT) training in improving peripheral intravenous cannulation (PIVC) skills.	Quiet Eye (QE) training helps to guide eye movement and focus during PIVC procedures. Compared to traditional training, it improves first-attempt success rates and can be integrated into nursing education to enhance skill development.	Future research should involve larger and more diverse groups, as well as long-term studies, to validate the training’s effectiveness and its impact on clinical practice over time. Additionally, incorporating QE-based PIVC training as a standard practice in nursing programs could be considered.
Vital and Nathanson (2023)/United States/a randomized prospective trial [9]	To evaluate the impact of the Stay S.A.F.E. intervention on nursing students’ ability to manage and respond to interruptions during medication administration.	1. Return to Task: The experimental group resumed tasks faster after interruptions. 2. Appropriate Response: They avoided unnecessary actions, unlike the control group. 3. Errors: Both groups made fewer errors over time; there was no major difference between them. 4. Task Load: The experimental group felt less frustrated and reported a lower workload than the control group.	The Stay S.A.F.E. strategy is suggested as a useful tool for teaching nursing students to manage interruptions effectively, helping them to prioritize tasks and reduce errors in high-interruption environments.
Cai et al. (2022)/Spain/a quantitative research design [16]	To examine the differences in visual attention between nursing students and expert practitioners when performing injection tasks.	1. Task time: Injection took less time than preparation. Experts worked faster in the second stage; students’ times stayed the same. 2. Eye tracking: - Focus: Experts looked at their thumb; students focused on fingers. - Eye movements: Experts had wider saccades during prep and smaller ones during injection; students’ eye patterns stayed steady. Gaze direction was similar, but patterns changed with each stage. 3. Errors: Both groups made fewer errors over time; there was no major difference between them. 4. Task load: The experimental group felt less frustrated and reported a lower workload than the control group.	Eye tracking technology should be considered as a tool to monitor and guide nursing students during training, facilitating adaptive learning experiences to enhance their skill development.
Sugimoto et al. (2022)/Japan/a quantitative research design [17]	To understand how experience influences situational awareness and visual focus during an intravenous (IV) injection task, using eye tracking to compare gaze patterns between experienced nurses and nursing students.	Nurses’ gaze patterns during IV injection tasks differed by experience. Experienced nurses focused more on key areas such as the patient’s face and IV devices, while students often missed steps such as adjusting drip speed. Using eye tracking and visual diagrams, the study showed that gaze behavior reflects situational awareness and errors. This approach may enhance nursing education, especially for complex, non-verbal tasks.	The study recommends using eye tracking technology to visualize differences in gaze patterns between experienced nurses and students. This method can improve nursing education by highlighting areas where students need more focus, especially for skills that are difficult to explain verbally.
Sanchez et al. (2019)/Canada/a quantitative research design [18]	To offer novel information about the understanding of eye behavior in human errors during handling of needles.	1. Task performance: Experts completed the injection task faster, while novices took longer, especially with scanning and injecting. 2. Accuracy: Both groups had similar accuracy, but novices were less consistent. 3. Gaze behavior: Novices focused more on the syringe and needle, while experts focused on the injection site, showing more efficient gaze patterns. 4. Attention management: Novices switched attention more often, whereas experts maintained focus longer, indicating better attention control and safer performance.	Eye tracking can help nursing students to improve their focus on key areas such as injection sites, rather than tools. The study recommends using eye tracking in nursing education, especially for procedures such as subcutaneous or intravenous injections. Future research should examine how gaze patterns change with practice and explore their role in injury prevention. Developing a self-learning system that lets students follow expert gaze patterns could support skill development and reflective learning. Further studies should assess the effectiveness of such a system.
Maekawa et al. (2016)/Japan/a quantitative research design [19]	To analyze the differences in eye tracking between skilled nurses and nursing students during intravenous injection procedures.	1. Nurses spent more time focusing on the puncture point, as it was crucial for inserting the needle accurately. 2. Nursing students had more dispersed gaze patterns, often looking at less relevant areas, such as articles and the patient’s face.	It is recommended to develop a self-learning support system that allows students to follow the gaze movements of experienced nurses for better image training and introspective learning. The quantification of gaze movements could be a useful method to enhance the development of nursing skills, particularly in the context of intravenous injection practice. Future studies should focus on evaluating the trial product of the proposed self-learning support system.
Amster et al. (2015)/United States/an observational study [20]	To investigate the reasons why nursing students fail to identify allergy errors during the medication administration process.	1. Error rate: 40% of students gave the contraindicated medication. 2. Students who noticed the error and those who did not had similar eye movement patterns. 3. The number of times participants looked at AOIs was not very different between the two groups. 4. The main reason for missing the error was a lack of pharmacological knowledge, such as not recognizing that amoxicillin is a type of penicillin.	It is recommended to enhance nursing students’ knowledge of medication classifications and commonly prescribed medications to reduce the risk of adverse drug events (ADEs). Additionally, eye tracking devices may be valuable for distinguishing between knowledge-based and rule-based errors, offering insights for targeted educational interventions.
Henneman et al. (2014)/United States/a quantitative and experimental design [21]	To compare verbal debriefing, eye tracking, and their combination to identify the most effective method for enhancing student knowledge and performance in patient safety.	1. All groups improved in post-tests compared to pre-tests. 2. The eye tracking group significantly improved in safety tasks such as checking patient IDs. 3. Combining eye tracking with verbal debriefing did not show any added benefit over eye tracking alone. 4. Eye tracking offered objective insights, especially in tasks needing visual and auditory checks. 5. The study supports eye tracking as a valuable tool for enhancing patient safety in simulation training.	The findings highlight the need for further research into effective debriefing methods post simulation, as little is known about the best way to provide feedback. Additionally, the use of eye tracking technology in simulations presents opportunities for both teaching and evaluation, as well as for studying healthcare practices that have not been explored before. Future studies should focus on identifying optimal visual scanning patterns that can help to reduce errors and improve patient safety outcomes.
Kataoka et al. (2011)/Japan/a quantitative study [22]	To examine changes in the visual behavior of nurses when operating an infusion pump under time pressure and dual tasking.	1. Time pressure (TP): Nurses worked faster with shorter eye fixations, but students showed inconsistent speed and focus. 2. Dual-tasking (DT): Increased mental workload; students and inexperienced nurses were slower and less focused, while experienced nurses remained stable. 3. Visual behavior: Time pressure reduced focus on key tasks, while dual-tasking shifted attention to less important areas, especially for students and inexperienced nurses. 4. Task difficulty: Experienced nurses found tasks easier, while students struggled, affecting their performance and mental workload.	1. Time pressure can help to shorten task duration, but may decrease the thoroughness of checking, particularly for those who are less experienced. 2. Dual-tasking can affect attention and task performance, especially when the nurse is less experienced, leading to possible errors. 3. Nurses with less experience need to be cautious under high-mental-workload conditions to avoid errors, suggesting the need for training or strategies to manage workload better.
Kataoka et al. (2008)/Japan/a quantitative research design [8]	To examine the visual behavior of nurses and nursing students during infusion pump operation under different conditions.	1. Visual behavior differences: Experienced nurses focused more on key areas such as the pump and tubing, while students looked at less relevant areas. 2. Impact of alarm sound: The air bubble alarm shifted attention to the pump and tubing as participants tried to fix the issue. 3. Fixation patterns: Eye movements and fixation times differed depending on clinical experience and the alarm sound.	1. Training: Increase practice to enhance students’ focus and performance during critical tasks. 2. Simulation learning: Use realistic scenarios (e.g., air bubble alarms) to strengthen clinical skills. 3. Critical focus: Teach students to prioritize key areas such as tubing and pump controls to avoid distractions.

## Data Availability

The data presented in this study are available upon request from the corresponding authors.

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
