# Peer review of "Evaluating Eye Tracking Technology in Nursing Education: A Scoping Review on Medication Administration Training"

_nursrep, 2025, doi:10.3390/nursrep15060185_

Round 1

Reviewer 1 Report

Comments and Suggestions for Authors

* Is the work a significant contribution to the field?

The focus topic is an innovative approach to determining medication errors during training. A suggestion for the authors: indicate what tool you are going to use to gather the data. It wasn’t until later that you mentioned smart glasses as one AI tool used to track eye movement.

* Is the work well organized and comprehensively described? *Is the work scientifically sound and not misleading?

Various concept analyses organized the work and detailed it. I don’t believe the work is misleading, but I do question the use of smart glasses or other tools to determine eye movement and how such behavior correlates with reducing medication errors.

A question beyond the scope of this work is whether universities with pharmacology courses lack the basic curriculum for medication application.  What or how are these courses taught such that an adjunct to learning may be necessary?  Just food for thought.

* Are there appropriate and adequate references to related and previous work?

Could the authors kindly review certain sentences for citation? . . . An integrative review of medication errors by nursing students during clinical practice identified six full-text articles and found that medication errors occurred at a rate of 1-6% (citation).

Did the authors include in their discussion answers to their research questions (section 2)? Would the authors consider adding the JBI methodology flowsheet and the PRISMA_ScR guidelines with indications of where those elements are located in the manuscript? These might be considered supplemental material and valuable for novice researchers interested in this topic.

Section 4.3 would work better in the introduction as a means of introducing the readers to the various instruments under consideration.

In section 4.4

. . . impact on clinical practice. Do you think the younger generation will become dependent on this technology rather than learning about medications?

. . . technical challenges, do you think age and economics play a role in this limitation?  

Section 5: discussion

. . . These studies focus on training and analyzing gaze patterns during important nursing tasks, provided valuable insights into skill development, attention control, and mistake prevention. Annotate this sentence at various points throughout the manuscript.

 Section 5.5

  . . .the results indicate. . . it is important to practice these procedures until the  brain has rote memory of the task. this cannot be stressed enough.

Section 5.6 Understanding how eye tracking works. . .  might be better to have in the introduction so readers know what you are talking about, how it works, what tools are used. . .etc.

Comments on the Quality of English Language

There are a few places where the past tense does not follow the present tense in the same sentence.  

Author Response

Thank you for your constructive feedback on our manuscript. We greatly appreciate the time and effort you invested in your review.

Please find attached a Word document detailing our responses to your comments, along with the amended version of the manuscript.

We hope the revisions address your concerns .

Thank you

Regrads,

Linda Ng

Reviewer 2 Report

Comments and Suggestions for Authors

Strengths:

    • Timely and original topic with high relevance to patient safety and simulation-based learning.
    • Methodologically rigorous approach following JBI and PRISMA-ScR guidelines.
    • Clear articulation of the review objectives and research questions.

Areas for Improvement:

    • Discussion structure: Consider merging or condensing overlapping subsections (e.g., 5.1 and 5.2), and providing more synthesis across findings.
    • Overgeneralization: Some recommendations (e.g., integrating eye-tracking into standard curricula) may be premature based on current evidence. Consider rephrasing these as suggestions for further exploration.
    • Limitations: The limitations section is appropriate but could benefit from acknowledging the potential for publication bias and the exclusion of non-English literature.
    • Figures and Appendices: Ensure the PRISMA flowchart and tables are properly formatted for final publication, and clearly referenced in the main text.

Language: Minor editorial polishing is suggested (e.g., subject-verb agreement, consistency in tense, sentence structure) but no major revisions are necessary.

Author Response

(The authors gave the same response as above.)

Reviewer 3 Report

Comments and Suggestions for Authors

Section 3.2, is it possible to add more detail or to describe the specific keyword/search phrase permutations used during the literature search to aid reproducibility.

The structure could be improved slightly, there are several tiny (1-2 sentence) sub-sub-sections in Section 3.3. I think this could be better presented/combined. 

3.3.2 what is the criteria for screening, please add more detail. 

Figure 1. in the screening section some of the lines for the arrows are missing or very thin, please fix these. 
-It would also be nice to see the number of records excluded by title/abstract and full text separately in figure 1.
-In figure 1 the box "Records excluded**" I can't see a matching ** footnote.

You may be interested in the following recent work, which combines eye tracking with head pose to provide a more complete understanding of focus and attention:

Khan, W., Topham, L., Alsmadi, H., Al Kafri, A. and Kolivand, H., 2024. Deep face profiler (DeFaP): Towards explicit, non-restrained, non-invasive, facial and gaze comprehension. Expert Systems With Applications, 254, p.124425.

Author Response

(The authors gave the same response as above.)
